# Predictors of Anxiety in Romanian Generation Z Teenagers

**DOI:** 10.3390/ijerph20064857

**Published:** 2023-03-09

**Authors:** Remus Runcan, Delia Nadolu, Gheorghe David

**Affiliations:** 1Faculty of Educational Science, Psychology and Social Work, “Aurel Vlaicu” University of Arad, 310032 Arad, Romania; 2The Department of Sociology, West University of Timisoara, 300223 Timisoara, Romania; 3Department of Agricultural Technologies, Faculty of Agriculture, University of Life Sciences “King Mihai I” from Timisoara, 300645 Timisoara, Romania

**Keywords:** anxiety, adolescence, predictors, Generation Z

## Abstract

Adolescence is a very complex period of life, full of challenges, and explorations that young people have to face on the path to becoming adults. In adolescence, specific deviations from the everyday lifestyle, as well as emotional failures or imbalances, may also occur. When things tend to become more and more unclear, adolescents come to directly face some form of anxiety. The present study concerns the relationship of Romanian adolescents with their fathers from the perspective of anxiety issues. For this, an anonymous questionnaire was applied using a sample of availability of 558 teenagers and a complementary second questionnaire was intended for their fathers (N2 = 114 subjects). The questionnaire addressed to Generation Z Romanian adolescents included items regarding the self-assessment of one’s own behaviour and relationship with one’s father, as well as the Generalized Anxiety Disorder Scale-7 (GAD-7). The questionnaire addressed to the fathers included mirror questions regarding the relationship with their children. The main results showed that the relationship between adolescents and their fathers has an ambivalent impact on anxiety: if it is a strong relationship, the risk to develop anxiety decreases, while if it is a weak relationship, it increases the risk of anxiety.

## 1. Introduction

According to Diagnostic and Statistical Manual of Mental Disorders: DSM-IV (1994) [1], there are 14 anxiety disorders, but only 5 develop in adolescents: generalized anxiety disorder, panic disorder, separation anxiety, social phobia (social anxiety disorder), and specific phobia [2]. Fromm-Reichmann suggested that anxiety is manifested by a frequently objectless feeling of uncertainty and helplessness, blocking of communication, intellectual and emotional preoccupation, and interference with thinking processes and concentration [3]. Anxiety is commonly associated with alexithymia [4], cyberbullying [5], depression [6], substance use [7], self-harm behaviours [8,9], and suicide [10].

Two aspects have been studied with predilection in relation to anxiety in adolescents: anxiety indices and anxiety predictors. A predictor is “a […] thing that predicts that something will happen in the future or will be a consequence of something” (Lexico). In adolescents with anxiety, the parent–adolescent relationship, use of the Internet, and time spent on electronic devices are the main predictors of anxiety [11,12,13].

Parent–adolescent relationship: According to Rosenberg, “the child who experiences prejudice is more likely to develop feelings of fear, anxiety, insecurity, and tension”, and they analyzed “the significance of family structure, sibling position, and parental behaviour for the child’s self-esteem and anxiety levels” [3]. Rosenberg seems to be the first to have analyzed the significance of parental behaviour (attention, desire for the occupational and social success of the child, disagreement, divorce, favouritism toward younger siblings, friendliness, indifference, interest, misjudgement of a child’s abilities, monetary support, reaction to a child’s friends or to school grades, remarriage, separation, supportiveness) for the child’s self-esteem and anxiety levels. Runcan and Drușcă investigated the father–daughter relationship and found that a father’s respect for his daughter prevents the latter from experiencing negative emotions such as anxiety, aggressive, and uncontrolled wrath [14]. Grant noted that the parent–child relationship is a “robust predictor of the development of anxiety disorders” and that parenting behaviours (control, lack of warmth, overprotection, rejection) were risk factors for anxiety disorders [15]. Parenting behaviour was also considered a unique predictor of anxious behaviour in adolescents [16] and an important extrinsic influence in etiological models of anxiety [17]. In addition, parental attitude and submissive behaviours of high school students’ parents predicted social anxiety at a significant level [18]. Time spent by parents with their children has an essential and well-defined role in the relationship between parent and child [19,20]. Last but not least, family climate also plays a major role in the emotional comfort of adolescents [21]. In the East European area, this climate has been increasingly affected in the last decade by the migration of one or both parents for lucrative activities in other countries [22].

Use of the Internet: Four decades after Rosenberg, other predictors of anxiety (information technology-related) came under scrutiny. Another predictor of anxiety in adolescents from Gen Z is the use of Internet [23]. Generation Z teenagers are native digital [24] and their everyday lifestyle is very much anchored in the online environment and in computer-mediated social interactions. Longitudinal associations of time spent doing Internet activities for communication purposes (i.e., instant messaging) versus time spent doing Internet activities for non-communication purposes (i.e., surfing) found that the latter predicted more depression and more social anxiety [25]. Ostovar et al. investigated the relationship of Internet addiction with anxiety, depression, loneliness, and stress in 1052 Iranian adolescents and young adults (16+ years) using the Depression Anxiety and Stress Scale (DASS), and found that Internet addiction is a predictor of anxiety, depression, loneliness, and stress, and that male Internet addicts differed significantly from females in terms of depression, anxiety, stress, and loneliness [26]. Runcan reached the same conclusion [27]. As for social media, Facebook use, which turns into Facebookmania in most cases [27,28,29], is considered the main predictor of social anxiety [30,31,32].

Time spent on electronic devices: The time spent using electronic devices is a significant predictor of symptoms of anxiety [33]. Mobile phones have become the media of choice for mediated interpersonal communication for adolescents, who treat them as companions with which they communicate, play, shop, and spend their leisure time, thus, threatening accepted norms of communication, cultural values, and sociability. Ha et al. evaluated the possible psychological problems related to excessive cellular phone use in adolescents and found that they expressed higher interpersonal anxiety, lower self-esteem, and more depressive symptoms [34]. Leung investigated addiction symptoms uniquely associated with mobile phone use (regarded as an impulse control disorder similar to pathological gambling) among adolescents in Hong Kong and identified four such symptoms: anxiety and craving, losing control and receiving complaints, productivity loss, and escape/withdrawal from problems. In the study, it was found that about 30% of adolescents use their mobile phone to communicate with their parents [35].

Of all the factors that can generate anxiety in adolescents, the relationship with parents and the family climate probably have the largest impact. In the context of an extremely dynamic and loaded daily agenda, one can assume that this factor is significantly altered by the contemporary lifestyle and, implicitly, the risk of anxiety in adolescents today is significantly higher. Moreover, the existence of a high level of digitalization and, implicitly, the quasi-total access to behavioral models, values, attitudes and individuals from significantly different socio-cultural areas only emphasize the problem. Even if positive parenting is a key element for the healthy development of children and adolescents [36,37], in Romania this is still in its early stages, with the school still holding control for the education and socialization of the young generation. On the other hand, anxiety is still viewed more as a specific passing state of age, and it is not given the attention it needs [38].

## 2. Materials and Methods

Between October and December 2020, a questionnaire for adolescents aged 13 to 21 from Romania was applied online, aiming to identify predictors of anxiety. The questionnaire was distributed on various virtual groups of teenagers and parents. Initially, only an availability sample was targeted, but the large number of registered responses (558), as well as the absence of any direction in the selection of subjects, allowed the assumption of a simple random representativeness for the population of teenagers with access to the Internet.

Five hundred and fifty-eight responses were recorded, allowing extrapolation of results for the digitized population in Romania with an error margin of 4.1%. It is worth mentioning that, in 2021, according to Romanian National Institute of Statistics data, 98.5% of people aged 16–34 frequently used the Internet. Although official data do not record the situation of Internet users under 16, it can easily be estimated that they retain a very high level of digital literacy. Consequently, the results obtained from this sample can be extrapolated to the entire population of adolescents in Romania, the share of those who do not constantly access the Internet being only 1.5%, i.e., below the calculated error margin. In order to ensure a better representativeness of the results, a share of the database on the sex variable at the level of adolescents aged 13 to 21 was also calculated: on 1 June 2021, national statistics recorded that 51.3% were boys and 48.7% girls.

The questionnaire addressed to adolescents was made up of coded questions: Q1.1 was developed by the authors, Q1.2–Q1.10 were from the Father–Daughter Relationship Rating Scale [39], Q1.11 was developed by the authors, Q1.12 was from the Generalized Anxiety Disorder Scale-7 [40], and Q1.13–Q1.19 were developed by the authors. In addition, exposure to risk factors (tobacco, alcohol, substances) was included in the analysis as direct questions (and, implicitly, without a control over the honesty of the answers) with a direct impact on the lifestyle of the adolescents. Due to the COVID-19 pandemic, the authors decided to shorten the reporting period from two years to one year. In the questionnaire addressed to adolescents’ fathers, Q1.1–Q2.5 were developed by the authors. All questions were agreed by the authors.

The questionnaire applied had an exploratory profile, aimed at assessing anxiety among adolescents in Romania during the pandemic and identifying the impact of its predictors: relation to father, exposure to risk factors, online behaviour, and socio-demographic variables. The application of a complementary questionnaire for the interviewed subjects’ fathers was attempted by generating a code for the adolescents at the end of their own questionnaire: fathers were supplied with this code together with a link for a questionnaire. Unfortunately, only 114 responses were obtained from adults, which drastically limited the achievement of in-depth comparisons. However, it was decided to preserve the questionnaires completed by fathers without postulating their representativeness. The questionnaires applied are shown in the Appendix A.

## 3. Results

The investigated sample was made up of 558 adolescents with the features shown in Table 1. For the construction of a regression pattern, the dependent variable was the score in the Generalized Anxiety Disorder Scale-7 (GAD-7) [40], while the independent variables were the father–adolescent relationship, exposure to risk factors, digital behavior, and socio-demographic variables (sex, age, residence, and perceived living standard). The descriptive results for each variable are presented below, as well as the way they aggregated.

Father–adolescent relationship: In general, the Romanian adolescents had a good relationship with their fathers, 77.6% stating a rather positive relationship (of different intensities, median = 5 out of 6). This was also reflected in the closeness with fathers, with 51.0% claiming they feel very close or extremely close (median = 3 out of 4) with their fathers. It was also notable that there was the perception of a highly positive attachment from the parent, with 65.0% of respondents considering that their fathers also enjoy much or very much their company (median = 3 out of 4). However, when the father is away from home, only 39.1% of adolescents (median = 2 out of 4) missed their fathers much and very much, either because their fathers’ absence is occasional or short-term, or because this issue is highly sensitive from the perspective of social desirability. For all these items, the only statistically significant difference depending on the sex variable was for the feeling of missing the father when he is away, which was stronger among adolescents (ANOVA test F = 3.994, sig. < 0.046, df = 1). All other distributions were not significantly different between groups.

In order to go into more depth with the analysis of the father–adolescent relationship, a series of items on the activities carried out together were included in the questionnaire. Thus, although 63.1% of adolescents enjoyed much or very much the time spent with their fathers, they believed that in the last year they had spent with their fathers a medium period of time (37.8%), the answers being relatively balanced on the 5-point response scale: 26.2% little or not at all, 36.0% much or very much. In the week previous to the survey, nearly half of the subjects (45.5%) indicated that they had spent less than one hour in daily discussion with their fathers (a total of 1–5 h for the entire week), while 14.3% said they had not discussed anything with their fathers at all. Similarly, activities shared weekly with their fathers in the last year had a balanced distribution: 17.5% of the subjects said they did not do anything together with their fathers in the previous week, 28.6% shared between 1 and 5 activities, and 53.9% shared at least 6 activities with their fathers during the last week (talking, watching movies, practicing sports, shopping, etc.).

When a new variable (v1 father) was generated by cumulating these items, the result was an aggregated image of the father–adolescent relationship, with values from 1 = a very bad relationship to 30 = an extremely good relationship (median = 19) (Figure 1).

The median value of these scores, 15, was taken as a scale cut-off point, and showed that 31.6% of adolescents have a deficient relationship with their fathers, while 68.4% have a positive relationship positive with their fathers. This variable was in a strongly significantly correlated (Table 2) with family communication (an item different from those included in its generation): Q12. How is, in general, your communication with your family members? (0 = extremely distant, 6 = extremely close).

For this variable (v1 father), there were no statistically significant differences between sexes (*t* test, t = −0.246, sig. < 0.806), both boys and girls recording a close average score of the father–adolescent relationship (18.1404 for boys and 18.2895 for girls). Instead, there was a significant difference for the age variable (ANOVA F = 3.042, sig. < 0.002, df = 2), 13-year-old subjects having a significantly higher average (22.9999) compared to 19-year-old subjects (15.3139). There was also a statistically significant difference for the variable material status (ANOVA F = 20.307, sig. < 0.0001, df = 3), those with a very good material state recording a higher score (21.1349) than those with a shortcoming in material status (13.2198). The same situation was also recorded for the residence variable (*t* test, F = 5.433, sig. < 0.020, t = −2.446, sig. < 0.15), people from rural areas recording a higher score (19.1818) than those from urban areas (17.6941).

Although mirror questionnaires being completed by only 114 fathers of adolescents participating in this study did not allow the extrapolation of results, a small descriptive analysis was conducted to highlight the correspondence between answers. Thus, in respect of the relationship with their own teenage children, fathers believed they have a very good relationship with them (median = 6 out of 6). Furthermore, the assessment of this relationship by both groups of subjects had a strong statistical correlation (Pearson correlation r = 0.607, sig. < 0.01) (Figure 2).

Exposure to risk factors: This was another set of questions concerning self-harm experiences and consumption of substances that prejudice health. A total of 79.9% never self-harmed, 70.4% never thought about committing suicide, 38.4% never consumed alcohol, 59.7% had never smoked, and 89.9% had never consumed drugs or other hallucinogenic substances. All these distributions were correlated statistically significantly, grouping two by two: self-employment (r = 0.530, sig. < 0.001), consumption of alcoholic beverages and smoking (r = 0.616, sig. < 0.001) and consumption of tobacco and hallucinogenic substances (r = 0.418, sig. < 0.001). A new variable (v2_risk) was generated by cumulating all the scores, and the following situation was presented: minimum = 5, i.e., no risk; maximum = 20, representing significant risks; median = 7 (Figure 3).

There was no significant difference for this variable v2_risk (*t*-test F = 1.305, sig. < 0.054, t = −0.515, sig. < 0.606), with both boys and girls recording close values (m = 8.2360, F = 8.3842). However, there were significant differences between ages (ANOVA F = 4.692, sig. < 0.0001, df = 8), with adolescents aged 13–14 years recording significantly lower values than those of older ones aged 20–21 years (13 years old = 6.9706 and 14 years old = 6.3871, compared to 20 years old = 9.0221 and 21 years old = 8.3212). Another significant difference was recorded in connection with material status (ANOVA test F = 5.246, sig. < 0.001, df = 3), those with a very good material status recording a lower score (7.5426) than those with a very poor material status (10.6331). Furthermore, significant differences were also recorded for the medium residence variable (*t* test, F = 7.298, sig. < 0.007, t = 3.666, sig. < 0.001), those from rural areas exhibiting less risks (7.6162) than those from urban areas (8.6788).

Of the fathers who responded to the questionnaire, 51.8% claimed they were much or very much concerned about their children’s activities and problems. This was also confirmed by a significantly statistic but inverse correlation between the v2_risk score and the father’s concern item (Pearson r = −0.223, sig. < 0.05). Thus, the higher the attention paid by the parent, the lower the cumulative value of the risk factors to which the teenager was voluntarily exposed. A similar correlation was also recorded between risk exposure and the knowledge of the adolescent entourage by the father (Pearson r = −0.296, sig. < 0.01), showing that parents who monitor their children’s time spent with their entourage group outside the family have a decreased risk exposure for adolescents.

Time spent on social media: As for digital behaviour, it was chosen to define it by time spent on social networks by adolescents. The answers to this question were differentiated depending on the sex variable (Figure 4).

For this variable, there were significant differences between the two sexes (ANOVA F = 27.917, sig. < 0.0001, df = 1), the girls tending to spend more time on social networks than boys (Pearson Chi-Square = 35.235, sig. < 0.001, df = 6). This variable exhibited a weak but significantly statistic correlation with the aggregate variable regarding the father–adolescent relationship (v1_father, r = −0.086, sig. = 0.041) and a strong positive statistically significant correlation with the risk variable (v2-Risk, r = 0.263, sig. < 0.001). Furthermore, the variable regarding the father–adolescent relationship showed a strong statistically significant but negative correlation with the risk variable (r = −0.363, sig. < 0.001). Notably, there was no statistic connection between time spent by adolescents on the Internet and the perception of their fathers on monitoring their children’s online activity (r = −0.036).

Anxiety: For anxiety analysis, we used the Generalized Anxiety Disorder Scale-7 (GAD-7) [40], which is composed of seven items on various feelings in the two previous weeks. The GAD-7 scale had a high level of reliability (Cronbach’s alpha = 0.862). The distribution of responses to these dimensions is presented in Table 3.

The distribution of summarized scores of this scale per type of anxiety is shown in Table 4.

There were significant differences between these scores depending on the sex variable (Pearson Chi-Square = 20.272, sig. < 0.001, df = 3), with girls showing a higher level of anxiety compared to boys. This confirmed the results of both Eisenberg et al., who found that “females were more than twice as likely as males to screen positive for anxiety disorders” [41], and Abbo et al., who found that the overall prevalence of anxiety disorders in their study was higher in females (29.7%) than in males (23.1%) [42]. The anxiety score was statistically significantly correlated with the three variables presented above: father–adolescent relationship, exposure to risks, and time spent on social media (Table 5).

When all these variables were included along with factual data in a multiple-linear-regression model, the following distribution was obtained (Figure 5 and Figure 6). However, the model had limited accuracy, as only 31.7% could be used for discussion and to draw conclusions. First of all, among the factors analyzed as independent variables, the highest impact on anxiety score was that of the cumulative variable of the risk behaviours (v2 risk), followed by age, sex, material status, time spent on social media, and father–adolescent relationship. It is of note that the environment had no contribution to the anxiety score (ANOVA F = 2.685, sig. = 0.102).

According to these distributions, high scores on the Generalized Anxiety Disorder Scale-7 (GAD-7) [40] were favoured by the adoption of risk factors in adolescent behaviour (self-harm, substance consumption), the existence of a dysfunctional relationship with their fathers, a precarious material status, and an extended time spent on social-media networks. Moreover, anxiety was more prominent at the onset of adolescence. Risk factors in adolescent behaviour (self-harm, substance consumption, suicidal ideation) can act as a major predictor for anxiety; they are also an effect of anxiety. The two variables are interdependent and influence each other—a high-level of anxiety generated by other causes could lead to the adoption of conduct in the risk factor area. For general purposes, we present the correlations between the variables used in Appendix A.

## 4. Discussion

Anxiety is a destabilizing factor of the emotional development of adolescents. Regardless of its causes, it cannot be ignored, especially when it tends to take severe forms. The father–adolescent relationship can have a direct impact on anxiety, either through avoiding it within the context of a solid relationship or, on the contrary, through its aggravation in the context of a deficient relationship. This relationship involves not only appropriately expressed and received empathy, but also the preservation of effective communication and, especially, the implementation of constant joint activities. Anxiety is directly correlated with the adoption of risk factors in adolescents’ behavior, self-harm and suicide attempts, and the consumption of harmful substances (tobacco, alcohol, drugs). When present, they act circularly, emphasizing each other: the appearance of slight anxiety favours risk behaviour, which does nothing but increase it. The vigilance of parents, and other adults vigilance is vital for the adoption of corrective measures in a timely manner, especially for the teenage debut period. Often, however, an episode of low-intensity anxiety passes unnoticed, in the context of age-affected family interactions or of a more precarious material status, which may also act as a possible predictor of anxiety.

## 5. Conclusions

In this contemporary hyper-digitalised society, in the context of social distancing imposed by the COVID-19 pandemic, another factor that can influence the onset of an anxiety episode is the time spent daily on social media. Escape in the digital universe is not only handy (the mobile phone being a gadget with which adolescents are already living in symbiosis), but also offers the mirage of a perfect social space, which is cosmetically enhanced with many suitable filters and extremely tempting, but at the same time can be extremely disappointing at the first concrete confrontation with a serious problem. Excessive social media use is already a scientifically verified source of depression, anxiety, alienation, and alteration of the harmonious emotional development of teenagers.

Our results confirmed those of similar previous studies [43,44,45,46,47,48,49,50] and highlight the importance of primary social support for Generation Z teenagers’ harmonious development. The limits of this research include the sample profile availability, in reservations concerning data generation, as well as the vulnerabilities inherent to the administration of an online questionnaire (i.e., the lack of control of the relationship with the respondents).

This study brings additional knowledge regarding the predictors of anxiety in Romanian teenagers in the context of the speedy digitization of daily life, the online school during the pandemic, a limited parenting system and, especially, the dynamic and fluctuating lifestyles of both young people and their parents.

Avoiding risk factors during adolescence and maintaining solid relationships with the family are two very important premises for overcoming emotional instability manifestations, which can be aggravated by chronic anxiety. The active and positive presence of the father—especially in early adolescence—is also a very useful resource for configuring the emotional and value profile of the future adult.

## Figures and Tables

**Figure 1 ijerph-20-04857-f001:**
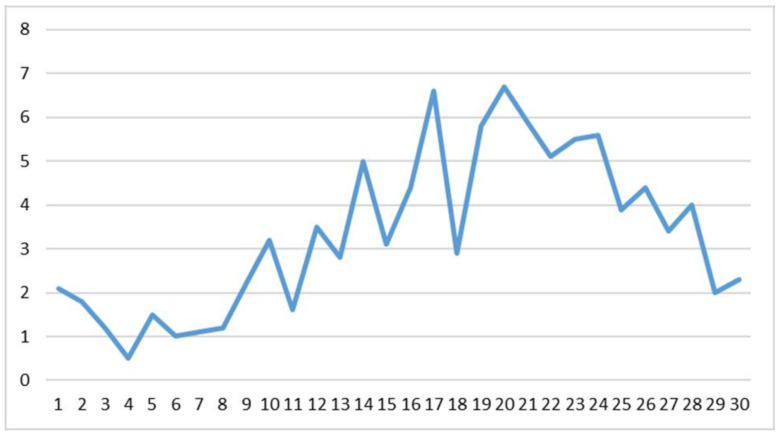
Father–adolescent relationship (v1 father) aggregate score.

**Figure 2 ijerph-20-04857-f002:**
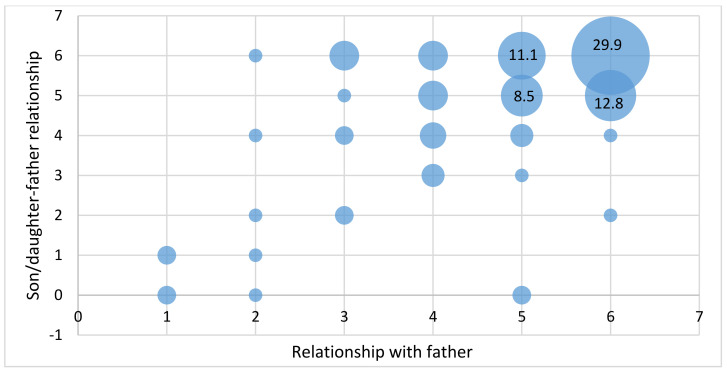
Assessment of father–adolescent relationship by both groups of subjects.

**Figure 3 ijerph-20-04857-f003:**
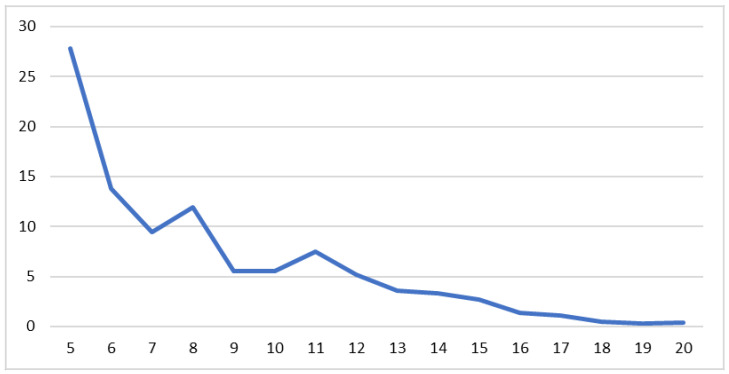
Risk aggregate score (v2 risk).

**Figure 4 ijerph-20-04857-f004:**
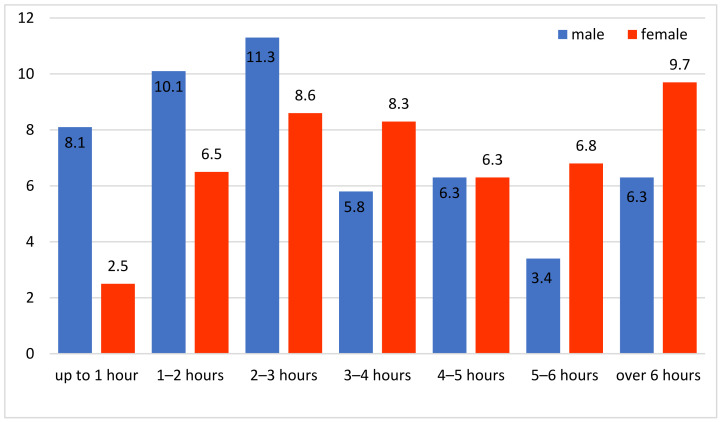
Time spent on social networks depending on sex (%).

**Figure 5 ijerph-20-04857-f005:**
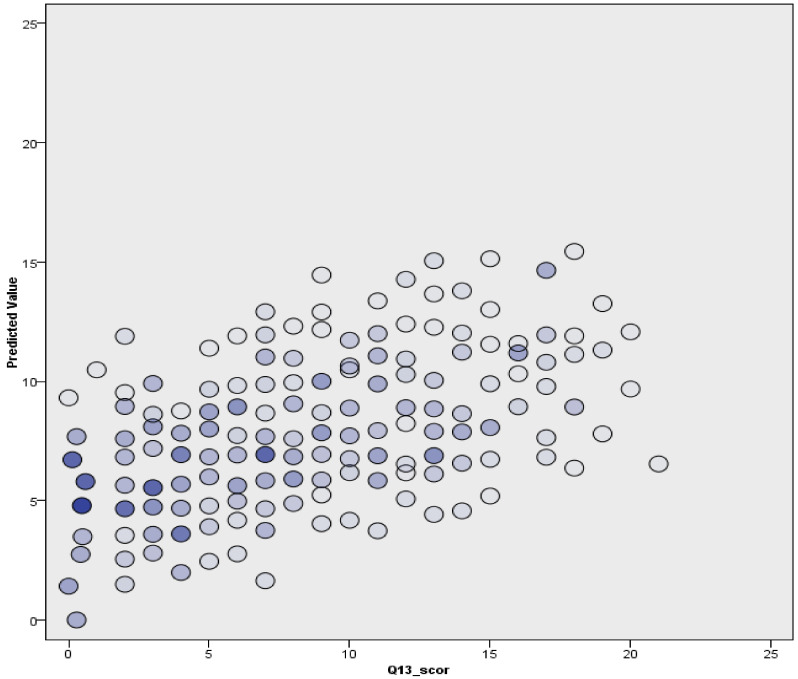
Multiple linear regression model.

**Figure 6 ijerph-20-04857-f006:**
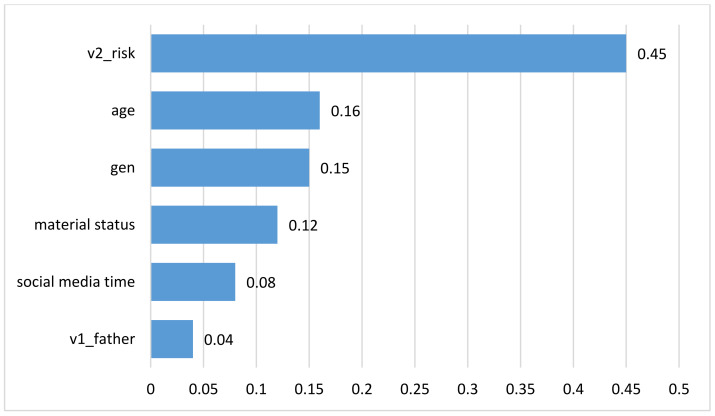
Independent variables.

**Table 1 ijerph-20-04857-t001:** Research sample (%).

Sex	Male				Female					
	31.9				68.1					
Age	13	14	15	16	17	18	19		20	21
(mean = 17.6 years)	4.3	7.5	15.8	24.7	9.3	7.2	9.9		10	11.3
Residence	Urban				Rural					
	64%				36%					
Material status	Very good		Good/comfortable		With certain shortcomings			Very bad		
	27.8		61.5		9.7			1.1		

**Table 2 ijerph-20-04857-t002:** Correlation of father–adolescent relationship with family communication.

		Communication with Mother	Communication with Father	Communication with Siblings
v1_father	Pearson Correlation	0.241 **	0.821 **	0.176 **
	sig. (2-tailed)	0.000	0.000	0.000
	N	558	558	558

** Correlation is significant at the 0.01 level (2-tailed).

**Table 3 ijerph-20-04857-t003:** The scores for Generalized Anxiety Disorder Scale-7 (GAD-7): Q13. In the past two weeks, how often have you been disturbed by any of the following problems?

	0 = Not at All	1 = Several Days	2 = More Than Half the Days	3 = Nearly Every Day
1. Feeling nervous, anxious, or on edge	25.5	35	17.1	22.3
2. Not being able to stop or control worrying	40.9	32.3	16.7	10.1
3. Worrying too much about different things	32.6	26.4	19	22.1
4. Trouble relaxing	46.3	23.4	15.8	14.5
5. Being so restless that it is hard to sit still	51.5	26.5	11.8	10.2
6. Becoming easily annoyed or irritable	26.1	32.4	22	19.4
7. Feeling afraid as if something awful might happen	48.4	25.2	12.2	14.2

**Table 4 ijerph-20-04857-t004:** The Generalized Anxiety Disorder Scale-7 (GAD-7) (Spitzer et al., 2006) [40].

Level of Anxiety Severity GAD-7 Scale Score	Frequency	Percentage
0–5—minimal	233	41.7
6–10—mild	154	27.6
11–15—moderate	108	19.4
15–21—severe	63	11.2
Total	558	100.0

**Table 5 ijerph-20-04857-t005:** Father–adolescent relationship, exposure to risk, and time spent on social media scores.

		v1_Father	v2_Risk	Time Spent on Social Media
Anxiety score	Pearson Correlation	−0.253 **	0.375 **	0.214 **
	sig. (2-tailed)	0.000	0.000	0.000
	N	558	558	558

** Correlation is significant at the 0.01 level (2-tailed).

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
