# Peer review of "Predictors of Anxiety in Romanian Generation Z Teenagers"

_ijerph, 2023, doi:10.3390/ijerph20064857_

Round 1

Reviewer 1 Report

Introduction

1. The introduction part is lack of theoretical base of the study. I am not clear what theory authors based on and how this research contributes to the extant literature on parent-adolescent relationship?

2. If the study is about father adolescent relationship, there is a lack of literature on how parenting practices affect such relationship. There are a lot of research in this area.

3. There is also a lack of introduction of Romanian young people in the background part. It could contribute to the international readers about the background information about Romanian adolescent.  For example, the prevalence of anxiety among them. Also international readers would be interested to know more about specific father adolescent relationship among Romanian.

Materials and Methods

4. This paper did not mention about how sample size was calculated.  It also did not mention how sampling procedure was done.  How the online questionnaire was distributed was not clearly mentioned.

5.There are some questionnaires developed by the author. However, there was nothing mentioned whether these questionnaires were based on some validated questionnaires or just developed by the author. What are the validity and reliability of the scale was not mentioned. Sample of the questions asked, and the scale used was not mentioned as well.

Results

6. In the results part, there is very little information presented about the background of the participants. Only age, sex residence and material status were mentioned. How about whether they are living with their parents? What is the marital status of their parents? Any special learning needs or disability among these adolescents? Nowadays we are in a world of very diverse family pattern. It worth mentioning all these for readers.

7. The age range of the participants was from 13 to 21 years old. It covers early adolescents to early adulthood. Any reference or theory that mention the appropriateness of doing a study with such large age range.

8. There was only more than a hundred mirror questions answered by father among the more than 500 questionnaires, any comparison was done between the background of these parents? It is also interesting to know the comparison of these answer with those which receive no father's response.

9.In the result part, it could be more helpful if reader can read the correlation matrix of all variables so that we can have an overall view of the study result.

Discussion

10. The discussion part is very brief and is not related to the existing literature. I suggest the discussion should directly relate to the findings that you have generated, and it should be discussed in accordance with theory or literature.  

11. This paper needs professional English editing before it can be published.

Author Response

Dear Reviewer

  1. The introduction part is lack of theoretical base of the study. I am not clear what theory authors based on and how this research contributes to the extant literature on parent-adolescent relationship?

Thank you very much for the observation, the parent-adolescent relationship was just one of the dimensions analyzed, our study targeting the predictors (in general) of anxiety in adolescents. But we have extended the subchapter dedicated to the parent-adolescent relationship (lines 43-69).

  1. If the study is about father adolescent relationship, there is a lack of literature on how parenting practices affect such relationship. There are a lot of research in this area.

The study also addressed the issue of parenting but did not focus on it because in Romania the phenomenon is still in its infancy.

  1. There is also a lack of introduction of Romanian young people in the background part. It could contribute to the international readers about the background information about Romanian adolescent. For example, the prevalence of anxiety among them. Also international readers would be interested to know more about specific father adolescent relationship among Romanian.

We added in as requested.

  1. This paper did not mention about how sample size was calculated. It also did not mention how sampling procedure was done. How the online questionnaire was distributed was not clearly mentioned.

We have added these details (lines 96-100).

5.There are some questionnaires developed by the author. However, there was nothing mentioned whether these questionnaires were based on some validated questionnaires or just developed by the author. What are the validity and reliability of the scale was not mentioned. Sample of the questions asked, and the scale used was not mentioned as well.

The additional questions included by the authors do not have the scale format (they are only a component of the sociological questionnaire). For the scale used, it has a high level of reliability (Cronbach's alpha = 0.862), as mentioned in the text.

  1. In the results part, there is very little information presented about the background of the participants. Only age, sex residence and material status were mentioned. How about whether they are living with their parents? What is the marital status of their parents? Any special learning needs or disability among these adolescents? Nowadays we are in a world of very diverse family pattern. It worth mentioning all these for readers.

Due to the subject we approached, we limited the factual data collected in order not to seem intrusive to the subjects. All study participants are enrolled in school and live at least with their father (some questions concerned weekly activities with the father). We did not include aspects related to special learning needs or disability because the approach itself was focused on general aspects. In Romanian family patterns are limited to monoparental, nuclear or extensive, so we have covered all by our approach.

  1. The age range of the participants was from 13 to 21 years old. It covers early adolescents to early adulthood. Any reference or theory that mention the appropriateness of doing a study with such large age range.

Indeed, we used a wider range of age because we followed the pre-marital period, when the relationship with the father is more consistent and with a greater impact on the development of young people. In Romania, 21 years is, generally, the age when higher education ends and, usually, this is also the time when most young adults leave their parents' homes and become autonomous altogether, especially due to employment.  It is a common practice to include, in studies on teenagers, young people aged 19-21 or more. Please find the following example, where the mean age is 19: Shane-Simpson, C., Manago, A., Gaggi, N. & Gillespie-Lynch, K. (2018). Why Do College Students Prefer Facebook, Twitter, or Instagram? Site Affordances, Tensions Between Privacy and Self-Expression, and Implications for Social Capital. Computers in Human Behavior, 86, 276-288. https://doi.org/10.1016/j.chb.2018.04.041.

  1. There was only more than a hundred mirror questions answered by father among the more than 500 questionnaires, any comparison was done between the background of these parents? It is also interesting to know the comparison of these answer with those which receive no father's response.

In order to pair the questionnaires, we asked the teenagers to establish a code and pass it on to their fathers. Unfortunately, a significant number of questionnaires could not be paired because they had the same codes (123456 or abcdef, etc.) and we had to give them up. For this reason, we did not do any analysis in detail between the teenagers whose parents agreed to answer the questionnaires and those who refused to do it.

9.In the results part, it could be more helpful if reader can read the correlation matrix of all variables so that we can have an overall view of the study result.

We have included in the initial text all statistically significant correlations, but not the insignificant ones. We have now added an appendix with the correlation table between all the variables.

  1. The discussion part is very brief and is not related to the existing literature. I suggest the discussion should directly relate to the findings that you have generated, and it should be discussed in accordance with theory or literature.

The discussion relate to the findings in literature is included as [43-50] in the Conclusions part.

  1. This paper needs professional English editing before it can be published.

Thank you very much, we have proofread the entire document.

Reviewer 2 Report

Thanks for giving me the opportunity to review this article. This is a meaningful study for us to understand the anxiety predictors among Z Generation adolescents. The topic is unique and important since the time is different and there are various influencing factors faced by adolescents with the development of technology. This manuscript is well-organized, but there are still some comments and suggestions that I might come up with for the authors’ further consideration:

1. There should be a citation of reference for the sentence “parent-adolescent relationship, use of the Internet and the time spent on electronic devices are the main predictors of anxiety” in the second paragraph. If the theoretical framework is developed from this point, then it should be elaborated for its source, development, previous studies, etc. Then authors should also explain why Z Generation adolescents’ anxiety predictors can be fitted into this theoretical framework.

2. It seems that previous studies have already identified the predictors of anxiety among adolescents, so it might be better to define research gaps for explaining why this study is going to study predictors of anxiety again. Maybe it is because of the specific context in Romania, or the target population is special, or there is something unique that previous studies haven’t focused on?

3. In the last paragraph of the Materials and Methods part, the authors reminded that they assessed the impact factors, including relation to father, exposure to risk factors, online behaviour, and socio-demographic variables. In the Introduction part, they only talked about parent-adolescent relationship, use of Internet and the time spent on electronic devices and considered they were the main predictors of anxiety, so please clarify why you chose to study “exposure to risk factors” as one of the predictors of anxiety in this study.

4. Some items in the questionnaires were made by authors, so they should be added data analysis to illustrate the internal insistency of the questionnaires in this study.

Author Response

Dear Reviewer 

  1. There should be a citation of reference for the sentence “parent-adolescent relationship, use of the Internet and the time spent on electronic devices are the main predictors of anxiety” in the second paragraph. If the theoretical framework is developed from this point, then it should be elaborated for its source, development, previous studies, etc. Then authors should also explain why Z Generation adolescents’ anxiety predictors can be fitted into this theoretical framework.

Thank you very much, we have included the references and developed the risks of digitalization for the development of anxieties in Z Generation.

  1. It seems that previous studies have already identified the predictors of anxiety among adolescents, so it might be better to define research gaps for explaining why this study is going to study predictors of anxiety again. Maybe it is because of the specific context in Romania, or the target population is special, or there is something unique that previous studies haven’t focused on?

We have added to the Conclusions the novelty elements brought by this study.

  1. In the last paragraph of the Materials and Methods part, the authors reminded that they assessed the impact factors, including relation to father, exposure to risk factors, online behaviour, and socio-demographic variables. In the Introduction part, they only talked about parent-adolescent relationship, use of Internet and the time spent on electronic devices and considered they were the main predictors of anxiety, so please clarify why you chose to study “exposure to risk factors” as one of the predictors of anxiety in this study.

We have added an explanation with the inclusion of the items about exposure to risk factors, more like factual variables because of the impossibility of controlling the veracity of the answers to these direct questions.

  1. Some items in the questionnaires were made by authors, so they should be added data analysis to illustrate the internal insistency of the questionnaires in this study.

The additional questions do not work as a distinct scale, they were rather aimed at the intensity of the relationship with the father on different components, such as: frequency of joint activities, common interests and concerns, affective support, etc. e have added Cronbach-Alpha index to the scales used.

Round 2

Reviewer 1 Report

I am satisfied with all the response and amendment you have made according to my comments in previous review. 

Reviewer 2 Report

Thanks for your revision. The topic is interesting and meaningful for understanding the mental health status among Generation Z teenagers. The manuscript is well-written and logic is clear, so it is about to be published.